# Friend or Foe: Protein Inhibitors of DNA Gyrase

**DOI:** 10.3390/biology13020084

**Published:** 2024-01-29

**Authors:** Shengfeng Ruan, Chih-Han Tu, Christina R. Bourne

**Affiliations:** Department of Chemistry and Biochemistry, University of Oklahoma, Norman, OK 73019, USA; shengfengruan@ou.edu (S.R.); chihhantu@ou.edu (C.-H.T.)

**Keywords:** antibiotics, DNA gyrase, GyrA, GyrB, quinolones, mobile genetic elements, target protection, gyrase inhibitors, gyrase regulation

## Abstract

**Simple Summary:**

DNA gyrase is an essential hub in bacteria that is targeted by small proteins found throughout the prokaryotic world. We explore the state of knowledge on potential roles for these gyrase inhibitors, including competition between mobile genetic elements and other cellular conflicts. Interestingly, a growing number of these inhibitors appear to protect gyrase by reversibly inhibiting its function. These intriguing mechanisms provide a glimpse into a complicated interplay of gyrase targeting and regulation. We anticipate that the underlying molecular mechanisms will be useful for antibacterial development in the face of increasing resistance to gyrase-targeting treatments.

**Abstract:**

DNA gyrase is essential for the successful replication of circular chromosomes, such as those found in most bacterial species, by relieving topological stressors associated with unwinding the double-stranded genetic material. This critical central role makes gyrase a valued target for antibacterial approaches, as exemplified by the highly successful fluoroquinolone class of antibiotics. It is reasonable that the activity of gyrase could be intrinsically regulated within cells, thereby helping to coordinate DNA replication with doubling times. Numerous proteins have been identified to exert inhibitory effects on DNA gyrase, although at lower doses, it can appear readily reversible and therefore may have regulatory value. Some of these, such as the small protein toxins found in plasmid-borne addiction modules, can promote cell death by inducing damage to DNA, resulting in an analogous outcome as quinolone antibiotics. Others, however, appear to transiently impact gyrase in a readily reversible and non-damaging mechanism, such as the plasmid-derived Qnr family of DNA-mimetic proteins. The current review examines the origins and known activities of protein inhibitors of gyrase and highlights opportunities to further exert control over bacterial growth by targeting this validated antibacterial target with novel molecular mechanisms. Furthermore, we are gaining new insights into fundamental regulatory strategies of gyrase that may prove important for understanding diverse growth strategies among different bacteria.

## 1. Introduction

Many essential bacterial pathways are known because of their targeting by molecular “warfare”, such as being in competition for resources between cells within bacterial cultures [1,2,3]. These strategies include secretion systems that introduce effector molecules into competitors, rendering the recipient cells inactive via interaction with essential targets [4,5]. Similarly, mobile genetic elements (MGEs) utilize strategies to target essential pathways to mediate retention within host cells (for recent reviews, please see [6,7,8,9]). MGEs are genetic material that can mobilize within communities including bacteriophage, plasmids, transposons, and even the recently discovered borgs [10]. In this framework, MGEs frequently target essential systems as a form of addiction, which is exemplified by toxin–antitoxin (TA) systems [11]. There are direct relations between MGE-mediated addiction and secretion system effectors [5,12], highlighting the common evolutionary origin converging on specific and essential molecular targets. It is interesting to note overlap between the targets of these systems and those of current use for antibacterial treatments, suggesting that mimicking warfare and addiction strategies would be fruitful to address the growing crisis surrounding a lack of new antibacterial development [13,14,15,16].

DNA gyrase is one of these essential and frequently targeted nodes both within bacterial systems and in antibacterial treatments, which is exemplified by the fluoroquinolone class of antibiotics [17,18]. Bacterial (circular) chromosomal replication is completely dependent on gyrase activity, while transcriptional homeostasis is also heavily impacted [19,20,21,22]. Given the primordial nature of many of these systems, it is logical that strategies to combat these warfare and addiction strategies have co-evolved. While antibacterial targeting relies on small molecules (for example, see [18]), our interest lies in the heavily under-explored molecular basis of protein-based gyrase inhibitors and their roles as either “friendly protector” or “toxic foe”. We believe this provides relevant insights into both the co-evolution of these strategies and to useful new approaches for gyrase inhibition.

## 2. DNA Gyrase Functions as an Essential Type II Topoisomerase in Bacteria

The catalytic cycle of DNA gyrase, shown in Figure 1, is distinct from other topoisomerases in at least two ways. Gyrase captures adjacent segments of DNA on the same molecule, and it interacts with DNA (“G” segment, Figure 1) using unique GyrA C-terminal domains (CTDs) (see the recommended literature [16,23,24]). Combining these features allows gyrase to introduce supercoils in the “negative” direction with different rates for the relaxation of positive supercoils versus the introduction of negative supercoils [25]. This is critical to relieve the torsional stress generated upstream of any biochemical function that requires parting a circular double-stranded DNA molecule, including DNA replication and RNA transcription [20,21]. While TopoIV is closely related to gyrase, it typically binds to segments of DNA from two separate molecules, such as removing the inter-linked circles generated by DNA replication, allowing separation into new daughter cells [26]. The catalytic cycle requires large conformational changes [19,27,28], and these intermediate interfaces are attractive targets for allosteric inhibition (see recent reviews [15,16]). This approach is exemplified by nature product inhibitors such as evybactin and albicidin and synthetic inhibitors such as the thiophene class [29,30,31].

While the catalytic functions of gyrase are conserved throughout bacteria, some sequence differences exist [22,32,33,34], and the rate of catalysis appears tailored to the doubling time of the specific bacteria [25,35]. For example, the gyrase from *E. coli* is among the most active, and it and its close homologs contain an extra insertion in GyrB though to enable faster turnover [32]. In contrast, the gyrase from *Mycobacterium tuberculosis* shares hallmarks with TopoIV, including an enhanced relaxation activity, and it is structurally diverse compared to other gyrase enzymes [36,37]. Overall, it is clear that the timing of gyrase activity is closely integrated into cellular homeostasis, making this an essential critical node throughout bacterial physiology.

## 3. Proteins Acting as Foes, Producing Damaging Gyrase Inhibition

MGEs are prevalent carriers of toxic proteins, such as gyrase-inhibiting toxins within toxin–antitoxin (TA) systems [11]. These impart an “addiction” phenotype by creating a dependency for cells to continually replenish a shorter half-life cognate antitoxin to maintain the neutralization of a toxin protein. TA systems are also abundant in bacterial and archaeal genomes [38]. It is unclear if gyrase-inhibiting toxins have roles in normal bacterial physiology, but they are typically not secreted from the producing cells. Some benefits have been attributed to specific TA systems, including the protection of gyrase from antibiotic inhibition [39,40] and cross-reactions that provide a “curing” or “anti-addiction” outcome [41,42].

### 3.1. ParE Toxin Proteins from ParDE Type II TA Systems

These genes were named based on initial observations of partitioning defects when they were deleted from the low copy number RK2 plasmid [43,44]. The subsequent discovery of ParE toxin inhibition of DNA gyrase can lead to confusion, as the gyrase-related Topoisomerase IV contains a “ParE” subunit with equivalency to DNA gyrase GyrB. Readers are forewarned, as functionally, the ParE toxin family and the ParE subunit of TopoIV are distinct and independent, and it is an unfortunate coincidence that they share the same gene and protein name.

Recent improvements to bioinformatics predictions indicate more than 4000 ParDE loci are contained within all sequenced prokaryotic genomes [38] with sequence homology typically below 30% [45,46]. The ParE subfamily can be distinguished from the related RelE-like toxins, however, by approx. 1.5 additional turns of a central helix–turn–helix motif (Figure 2). ParDE TA systems are enriched in mobile genetic elements, including plasmids carrying resistance genes, where they function as canonical addiction modules [38,39]. Many chromosomal systems, however, are within integrated phage or pathogenicity islands, which is consistent with their capture from previously mobile genetic elements [47,48,49,50,51,52]. 

ParE toxins were suspected to target DNA gyrase based on early observations of expression triggering a filamentous phenotype [53]. This has subsequently been confirmed in addition to also inducing chromosomal aberrations and a corresponding “SOS” repair response indicative of the accumulation of DNA breaks [40,47,49,53,54,55,56,57,58,59]. The expression of ParE toxins has been noted to provide target protection to anti-gyrase and other antibiotics [39,40], although it seems unlikely that such a toxic inhibition would serve a protective function.

The documentation of in vitro gyrase inhibition is consistent with these phenotypic results with a persistence of dsDNA breaks and IC_50_ values in the low micromolar range [40,54,55,57]. ParE toxins do not inhibit topoisomerase IV nor the relaxation reactions mediated by gyrase ([54,55] and unpublished observation). The inhibition of gyrase by ParE toxins requires ATP hydrolysis [54,55], indicating that ParE toxins likely recognize and stabilize a catalytic intermediate adopted after dsDNA cleavage and upper strand transfer but before DNA re-ligation can occur. However, alternative models of the gyrase catalytic cycle raise questions about the specific ordering of ATP hydrolysis and subsequent resolution of the DNA break that may be relevant to this inhibition [60,61].

The specific molecular mechanism of ParE-mediated gyrase inhibition remains unclear, but gyrase bearing mutations to the small molecule inhibitor nalidixic acid or to the CcdB toxin can still be inhibited by ParE [55]. A ParE toxin encoded on chromosome II of *V. cholerae* bound tightly to *E. coli* GyrA59 with biphasic kinetics and nanomolar or better K_D_ values [55]. A ParE toxin encoded within a prophage region specific to the *E. coli* O157:H7 strain did not bind to this GyrA59 region and was published as not inhibiting gyrase [62]. In contrast, a ParE encoded in *M. tuberculosis* bound only to the GyrB subunit of Mt gyrase with a K_D_ that is mid-micromolar [57]. Our own studies have measured an even weaker interaction of an attenuated ParE toxin encoded in the *P. aeruginosa* genome to a fused *E. coli* GyrB-GyrA construct (unpublished observation) with weaker IC_50_ values of this ParE for its host gyrase versus the from *E. coli* [40].

Multiple studies have reported that C-terminal truncations of approx. 10 amino acids from ParE toxins, or mutating specific amino acids in this region, reduce phenotypic toxicity [56,63,64]. An attenuated ParE toxin with weak toxicity has a shortened C-terminus, giving support to the toxicity of ParE proteins residing within their C-terminal regions [65]. All of the available crystal structures of ParE toxins are paired with their cognate ParD antitoxin, and the C-terminus of ParE toxins interacts with this neutralization partner; in addition, most of the terminal amino acids of ParE are frequently disordered (indicated by “Δ” notations on Figure 2). However, AlphaFold models (Figure 2, gray) suggest this C-terminal region may adopt a helical conformation. Of note, two amino acids required for the toxicity of the *M. tuberculosis* ParE2 [63] are positioned on this helix, and the AlphaFold model places them interacting with an absolutely conserved Asp amino acid in the first helix. Elusive structural details of interaction with gyrase, paired with conflicting results for binding, makes the molecular mechanism of ParE toxins a major gap within gyrase inhibition profiles.

**Figure 2 biology-13-00084-f002:**
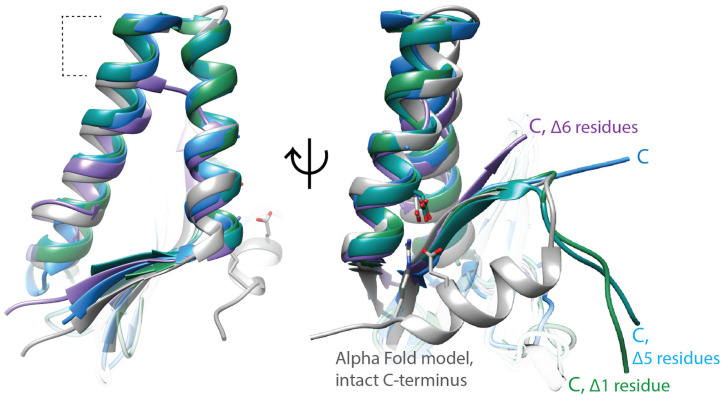
ParE toxins have canonical structural features, but the molecular mechanism of gyrase inhibition remains unclear. ParE toxins can be distinguished from other super-family members by the extra turns in the helix–turn–helix motif (dashed lines, left panel). The toxicity of ParE proteins appears dependent on the C-terminus. C-terminal amino acids too disordered to place in the crystal structures are indicated with “Δ”. Note the ParE shown in blue has a shorter C-terminus and has been published as having attenuated toxicity. *Superposed structures:* PDB 3G5O, *M. tuberculosis* RelE2 [66], purple; PDB 3KXE, *Caulobacter crescentus* ParE1 [67], green; PDB 6XRW, *P. aeruginosa* ParE1 [65], blue; PDB 7R5A, *V. cholerae* ParE2 (Garcia-Rodriguez et al., unpublished), dark cyan; *M. tuberculosis* ParE2 Alpha Fold model (AF-P9WHG5-F1_v4), dark grey.

### 3.2. CcdB Toxin Proteins from CcdAB Type II TA Systems

Early studies identified the CcdBA system encoded on the F plasmid and that the target of the CcdB toxin is DNA gyrase [68,69]. Upon CcdB expression, the resulting phenotype resembles that of quinolone and of ParE toxin inhibition, including the accumulation of dsDNA breaks, a triggering of the SOS response, and filamentation phenotypes. The latest bioinformatic analysis has identified around 3800 CcdBA systems, locating primarily on plasmids, although integrated genomic versions are found in some bacteria [38]. There is some evidence to support a role of both plasmid and integrated TA systems as competing with other MGEs, providing a protective advantage to host cells from invading genetic material [41,42,70] with specific examples for this TA system [63,71]. The strong toxicity mediated by CcdB has been co-opted for use in multiple biotechnology applications including plasmid selection and screening for CRISPR activity [72,73,74,75]. CcdB has also been formulated into a potent treatment by combining the toxin with intein sites, allowing controllable targeting of the CcdB activity in selected bacterial cells [76] or in other approaches by the direct administration of a peptide derived from the CcdB sequence [77]. 

Studies of the molecular mechanisms of gyrase inhibition have focused on the CcdB encoded on the F plasmid, the chromosomally integrated toxin in *E. coli* O157, and the chromosomal version from *Vibrio fischeri* [42,63,78,79]. Early studies selected for gyrase mutants resistant to the F-plasmid CcdB toxin and that identified a single change in GyrA Arg462 to Cys (Figure 3) was sufficient to protect cells from CcdB toxicity [80]. While CcdB can bind to both isolated GyrA and within a gyrase–DNA complex [80], the physiologically relevant inhibition appears to only occur during transit of the upper “T” DNA strand through the broken “G” segment [81]. This conformational specificity arises from conformational changes in GyrA, including a significant steric hindrance between the upper central portion of GyrA and CcdB until strand passage triggers this middle “gate” to open (Figure 3, yellow asterisk) [82]. When CcdB binds to gyrase, the catalytic cycle is stopped before ligation can occur (Figure 1, step 4), resulting in the accumulation of dsDNA breaks [81]. 

While there is some sequence variability between CcdB toxins and their contacts with gyrase from different bacterial species, the gyrase binding site of CcdB is consistent and serves to prevent the bottom gate of gyrase from opening to release the captured DNA strands (Figure 3) [78,82,83]. Despite sequence differences, the overall binding affinity of CcdB toxins for GyrA appears in the low nanomolar range [82,83]. A chromosomal version in O157 strains of *E. coli*, however, appears somewhat attenuated with respect to toxicity and can protect cells from the F-plasmid version of CcdB by neutralization via cross-interaction with the chromosomally encoded CcdA antitoxin [63,70].

Earlier studies identified peptides based on the CcdB toxin that were able to mediate both gyrase and topoIV inhibition [84]. The recent success of this antibacterial approach has been reported for a 24 amino acid peptide mimetic of CcdB appended with membrane-penetrating and stabilizing modifications (corresponding to the yellow helix on CcdB molecules; see the inset in Figure 3) [77]. This peptide had strong therapeutic efficacy in mouse models of infection with *Salmonella typhimurium, Staphylococcal aureus,* and *Acinetocabacter baumannii.* These exciting results identified that while a mutation of GyrA rendered it less susceptible to the CcdB-corresponding peptide, no escape mutants were generated for TopoIV inhibition [77]. The affinity of the peptide for the ParC subunit of TopoIV (equivalent to GyrA) was approx. 1 nM, while for binding to the GyrA fragment “14” (corresponding to the inset structures, Figure 3), it was lower at approx. 51 nM [77]. The peptide affinity for GyrA-14 was noted to be approx. 10-fold weaker than that previously measured for the full CcdB toxin [77,85,86], while CcdB binding to GyrA (the “59” central portion) was approx. 0.5 nM [86].

**Figure 3 biology-13-00084-f003:**
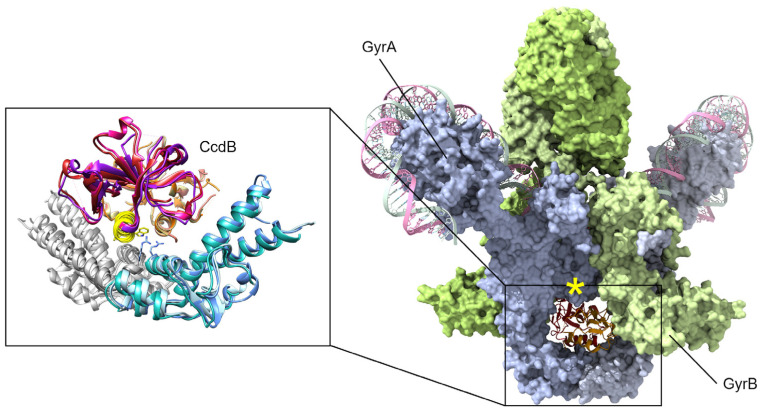
CcdB toxins are dimeric and bind to the bottom “gate” in GyrA and block the release of either DNA segment (only the cleaved segment is shown, “G” in Figure 1), resulting in an accumulation of dsDNA breaks. Binding can only occur when gyrase is midway through the catalytic cycle, as GyrA dimers open at the site of dsDNA cleavage to allow transit of the top “T” segment of DNA (not shown here, see Figure 1). A peptide mimic, corresponding to the yellow helix (insert) of CcdB, has shown antibacterial efficacy in a mouse infection model [77]. Mutations at gyrase Arg462 (shown as sticks in the inset) impart resistance to CcdB toxicity. In available holo-gyrase structures, all GyrA are in the “closed” conformation correlating to the conformational state before strand passage. When the full gyrase structure is superposed with the CcdB structures (right, GyrA blue surface, GyrB green surface), a steric clash is evident (marked by a yellow asterisk). *Superposed structures:* PDB 1X75, *E. coli* GyrA and F plasmid CcdB [82], PDB 4ELY, *Shigella flexneri* GyrA and *V. fischeri* CcdB, and PDB 4ELZ, *V. fischeri* GyrA and CcdB [78]; these are superposed onto PDB 6RKW, *E. coli* GyrA and GyrB bound to a 130 bp DNA [87].

### 3.3. TsbT Toxin Proteins from TsbAT Type II TA Systems

In 2019, Kato et al. first demonstrated four candidates of two-gene operons worked as TA systems in Staphylococcus aureus strain N315, including the TsbAT system [88]. Later studies demonstrated the target of the TsbT toxin was DNA gyrase, as expression within *E. coli* or *S. aureus* produced increasingly relaxed chromosomal DNA, and in vitro assays with purified gyrase mixtures were also potently inhibited [89]. TsbT-mediated inhibition appears bacteriostatic [88], and homologs from other *Staphylococcus* species have varied toxicity that appears to correlate with differences in primary sequences [89]. Assays monitoring in vitro gyrase activity demonstrate inhibition after DNA breakage but before re-ligation, resulting in the accumulation of double-stranded DNA breaks [89].

To date, there are no structures for this approx. 50 amino acid protein, and studies are relatively limited. Site-directed mutagenesis revealed that residues E27 and D37 are critical for toxicity (Figure 4) [89]. Secondary structure analysis suggests that these residues are likely within an alpha-helix, while the N- and C-termini of the small protein appear unstructured [89]. Interestingly, the highly polar sequence within this helical region has striking similarities to the second alpha-helix of some ParE toxins (Figure 4, blue ribbon), suggesting they may have an overlap in the molecular mechanism of toxicity.

### 3.4. Microcin B17

Microcin B17 (MccB17) is a glycine-rich “mini” colicin peptide carried on single-copy plasmids in *Enterobacteriaceae* that can be excreted by producing cells [90,91]. MccB17 is produced when growth conditions become limiting, inducing bacterial stress, which is consistent with a role in limiting competition for resources by targeting neighboring bacteria with this colicin [92,93]. The encoding operon [94] includes an immunity protein with high similarity to pentapeptide repeat protein MfpA (see below) to protect the producing cell [95] and export machinery [96]; in addition, the operon also encodes a synthetase complex required for post-translational modifications [97]. The toxic activity of MccB17 requires these post-translational modifications to introduce oxazole and thiazole heterocycles of serine and cysteine precursor side chains. The final product MccB17 is generated by removing the leader peptide in the N-terminal region by an endogenous protease [98,99]. While the *E. coli* derived MccB17 is a weak gyrase inhibitor, related microcins produced by Pseudomonads may instead function as translation inhibitors [100,101].

The molecular mechanism of MccB17 interactions with gyrase remains unknown. Mutation of Trp751 in GyrB can impart modest resistance [102]. Studies have identified that MccB17 blocks the transport of the uncleaved DNA segment, potentially requiring a sufficient length of DNA to first mediate cleavage as well as ATP hydrolysis [103,104]. Interestingly, the GyrA CTDs that position DNA for the double-strand cleavage were found to be dispensable [105]. Consistent with this, MccB17 can also inhibit gyrase relaxation activity, which does not rely on the GyrA CTDs [105]. The relaxation of supercoiling by gyrase has a low intrinsic rate and typically occurs when the two segments of captured dsDNA are not closely positioned on the same molecule [25]. It is generally agreed that MccB17 is a weak inhibitor of both gyrase supercoiling and relaxation reactions, likely recognizing an intermediate in the catalytic pathway after dsDNA cleavage but before transit of the intact segment through this break [14,104,105,106]. The resulting accumulation of dsDNA breaks results in toxicity that slows the growth of potential competitor bacteria that lack a protective mechanism.

## 4. Proteins Acting as Friends, Protecting Gyrase from Damaging Inhibition

Other gyrase-inhibiting peptides have been identified in the genomes of bacteriophage. These are an interesting category, as they appear to limit host cell replication to favor phage production. This suggests a regulatory role, although the modes of inhibition are themselves less damaging. Similarly, some post-translational modifications appear to inhibit gyrase in a readily reversible manner, seeming to protect cells from damaging gyrase inhibition. Other proteins have “moonlighting” functions that may link gyrase activity to other replication events, thereby serving a regulatory function.

### 4.1. Phage-Derived Peptides

The interplay of bacteriophages with bacterial cells is complex and can reveal unique interactions. Two bacteriophages, one integrated into genomes of *Corynybacterium glutamicum,* and the other infecting *Pseudomonas aeruginosa*, each encode a small peptide that regulates gyrase activity [107,108]. Reports on these peptide gyrase inhibitors are recent and, as of yet, limited. It seems reasonable, however, that similar peptides remain unannotated as significant regions of phage genomes contain uncharacterized small proteins [109].

The *C. glutamicum*-specific ~50 amino acid peptide (Cg1978) was named Gip, for gyrase-inhibiting-peptide, and it is encoded in the CPG3 prophage region of the bacterial genome [108]. Upon prophage activation, the peptide appears to limit host cell replication by reversibly inhibiting gyrase supercoiling. The overexpression of Gip triggers prolonged lag phase growth, and it is suspected that subsequent phage induction may be linked to changes in host genome topology that arise through actions of Gip. While there was no accumulation of dsDNA breaks, an SOS response was initiated; however, this response could arise from other concomitant cellular events.

Co-precipitation assays identified GyrA as the main interaction site, and a K_D_ value of 5 nM was determined by surface plasmon resonance [108]. Activity assays revealed a significant specificity for the Corynebacterial gyrase with a half-maximal inhibition of approx. 5 µM, while that for Mycobacterial gyrase was 30 µM, and the more divergent *E. coli* gyrase was only modestly inhibited even at 60 µM [108].

Similarly, among the *Bruynoghevirus* genus of *Pseudomonas* phage, there are homologs of ~50 amino acid peptide gp9 derived from the LUZ24 phage, which is now classified as an inhibitor of gyrase and renamed “Igy” [107]. While its expression did not impact *E. coli* cells, within *P. aeruginosa* strains PAO1 and PA14, a filamented phenotype was induced that was consistent with damaging gyrase inhibition. In vitro assays revealed a half-maximal inhibition against *P. aeruginosa* gyrase supercoiling of <12.5 µM with no impacts on gyrase relaxation, and they highlighted a lack of linear DNA suggesting the peptide is not functioning as a gyrase poison [107]. Consistent with this, ciprofloxacin-resistant gyrase was still inhibited by Igy. Interestingly, it appears that Igy was not inhibitory toward gyrase when it carried an appended N-fusion protein (MBP), suggesting steric hindrance at this surface. A predicted model for Igy was used to support suggestions of functioning as a DNA mimic, whereby a negatively charged surface may interact with GyrB and thus block DNA binding [107]. Two-point mutations on Igy, a glutamic acid and a threonine, were selected in screens for loss of inhibitory activity.

While the specific molecular mechanisms appear to differ between these two phage peptides, both appear to specifically target the host gyrase enzyme. Furthermore, they both reduce gyrase activity but without inducing a damaging SOS response, suggesting a role in altering the host cell replication to favor phage production.

### 4.2. FicT Toxins from FicTA Type II TA Systems

“Filamentation induced by cAMP”, or a fic phenotype, is a filamentous form of growth first noted for *E. coli* PA3092 when grown at high temperature and in the presence of cyclic AMP [110]. Kawamukai et al. later revealed that the *fic-1* gene product (Fic protein) is responsible for the induction of this phenotype [111]. Fic proteins catalyze the transfer of adenosine 5′-monophosphate moieties onto target proteins, typically modifying a Tyr side chain by AMPylation (also called adenylylation). Fic proteins are widespread and share a common active site sequence (HxFx[D/E]GNGRxxR) and catalytic activity, but they are diverse with regard to substrates that are modified, seeming to function in regulatory mechanisms as well as toxicity [112,113,114]. 

The FicTA module is a novel Fic-related type-II TA system [112]. The ectopic expression of the *Bartonella schoenbuchensis* Fic protein VbhT, the YeFicT of *Yersinia enterocolitica* strain 8081, and EcFicT of *E. coli* K-12 were each shown to result in strong inhibition of *E. coli* growth [112,115]. Using mass spectrometry, Harms et al. showed that VbhT adenylylated *E. coli* GyrB at Tyr109 and TopoIV at Tyr105, causing an inhibition of the ATPase activity of these homologous protein domains. The AMPylation of GyrB and TopoIV-ParE has been noted to block growth by halting replication [112,116,117]; however, no dsDNA breaks are generated. Other TA system toxins, such as the Doc toxin, are also Fic-type AMPylators, but their targets are in pathways of translation [118]. In general, Fic proteins typically recognize their specific targets using a conserved beta-hairpin motif; however, the specific recognition elements on the diverse Fic targets remain unclear [114]. The *Bartonella* FicT has been demonstrated to be part of a Type IV secretion system, and links between integrated conjugative plasmids have been identified [5], suggesting that topoisomerase inhibition may help regulate conjugative plasmid replication or transfer.

### 4.3. Pentapeptide Repeat Proteins

Pentapeptide repeat proteins share a three-dimensional structure that resembles a beta solenoid fold, generating a repeating pattern that can mimic a helical structure such as DNA. The naming of this fold arises from a regular repeating sequence of five amino acids, which are arranged with regularity that projects the first and the third amino acid inward with the remaining second, fourth, and fifth amino acid side chains pointing outward and typically displaying a negative character. These repeating units of five amino acids are arranged along the primary sequence to be linked into four regions of sequence variation, giving rise to one “turn” of the solenoid. Some proteins with this fold contain one or two larger loops that protrude from one side of the solenoid and impact the binding interactions with biological partners. These solenoid folds are typically found as dimers that interact via C-terminal helices, such that each free end of the solenoid is an N-terminus of the pentapeptide repeat protein. Among this protein class, the Qnr and MfpA proteins are known to interact with gyrase and produce a beneficial phenotype to the host cells.

#### 4.3.1. Qnr Proteins

The first identified Qnr was encoded on the pMG252 multi-resistance plasmid harbored in a clinical isolate of *Klebsiella pneumonia* [119]. Homologs of Qnr have subsequently been identified on other resistance-bearing plasmids and in the chromosomes of several Gram-positive and Gram-negative bacteria [120]. A classification system distinguishes variants based on the sequences of Qnr proteins, where different Qnr classes have up to 65% sequence similarity, while members within classes typically vary no more than 10% in primary sequence [121]. In all, there are six classes in current databases with the QnrA class containing the originally identified sequence, QnrB, QnrC, QnrD, and QnrS enriched on plasmids and chromosomes, and QnrVC harbored within the chromosome of *Vibrio cholerae* [120,122].

Studies have shown that both type II topoisomerases, gyrase and TopoIV, are protected from quinolone inhibition in the presence of Qnr proteins [119,120,123]. However, Qnr proteins do not protect from the GyrA-targeting simocyclinone D8 compound that affects DNA binding; moreover, Qnr did not protect from either synthetic or non-synthetic GyrB inhibitors [123]. Taken together, Qnr proteins appear to be protective only against compounds that stabilize the cleaved state of the DNA–gyrase complex.

Qnr proteins were demonstrated to bind to both individual GyrA and GyrB subunits, and to the holoenzyme, by both native gel shifts and by bacterial two-hybrid assays; of note, there was no requirement for DNA or gyrase-inhibiting compounds such as quinolones for binding [124,125]. Further evidence suggests that Qnr proteins are able to inhibit gyrase supercoiling activity but only poorly inhibit relaxation activity [126]. Expression of the EfsQnr protein in *E. coli* cells did not impact division but did block separation, forming clusters of daughter cells [126].

Sequencing data confirm that Qnr proteins predate antibiotic use, and plasmidic versions are derived from bacterial chromosomes [121]. This suggests that bacteria had use of Qnr proteins to protect topoisomerases throughout evolution, although endogenous functions have not been determined. The first description of *qnr* genes is from the 1930s, which predates quinolone use by at least 30 years [127]. Alternative endogenous functions are also suggested by the ubiquity of *qnr* genes among various bacterial hosts. One plausible suggestion is the functioning of Qnr proteins in response to DNA damage. This suggestion is supported by discoveries of a LexA-binding site in the promoter region of *qnrB* alleles, implying that the expression of QnrB might be regulated by the SOS response known to signal DNA damage [128]. Qnr proteins might also play an adaptive role for bacteria to survive in a stringent environment by altering gyrase activity and thus supercoiling of the genome. Kim et al. showed that *qnrA2* gene is overexpressed with cold shock in *Shewanella algae* and suggested this helps reprogram expression profiles to favor survival [129]. Overall, Qnr proteins appear to both regulate gyrase activity while also potentially promoting a mutagenic phenotype via SOS activation, overall imparting beneficial traits to the host bacterial cell.

#### 4.3.2. MfpA Proteins

Mycobacterial fluoroquinolone resistance protein A (MfpA) is a pentapeptide repeat protein that was first identified by genomic screening in *M. smegmatis* [95]. It has since been discovered in chromosomes of at least 19 species of *Mycobacterium* as well as at least 10 *Actinobacteria* species, all with approx. 40% homology [130]. The structure of MfpA proteins are beta solenoids with polar side-chain distributions that mimic double-stranded DNA [131,132]. MfpA can bind to DNA gyrase but inhibits supercoiling poorly [131,133]. Interestingly, homologs from *M. tuberculosis* (MfpA_Mt_) and *M. smegmatis* (MfpA_Ms_) share 67% identity, but MfpA_Mt_ requires an accessory protein, MfpB, a small GTPase, to protect gyrase from *M. tuberculosis* from quinolone inhibition. In contrast, MfpA_Ms_ does not require any accessory proteins to protect gyrase from quinolone-induced DNA cleavage by gyrase [131,133,134].

The MfpA_Ms_ has been proposed to interact with the ATPase domain of GyrB, as it is unable to protect gyrase in the absence of ATP [133]. Furthermore, ATP hydrolysis was also required, and MfpA_Ms_ was also shown to activate the ATPase activity of Mycobacterial gyrase while competing with the binding of linear DNA [133]. A crystal structure of MfpA_Ms_ complexed with a domain of *M. smegmatis* GyrB containing the ATPase domain highlighted an interface critical for their interaction as well as the protection from FQs [133]. This led to the suggestion that MfpA_Ms_ acts as a DNA T-segment mimic (see Figure 5) that also stimulates ATP hydrolysis. The activated ATPase region subsequently triggers a conformational change leading to the opening of the blocked DNA gate, thereby expelling bound fluoroquinolones and allowing relegation of the broken DNA segment [133]. This elegant strategy thereby can protect but can also rejuvenate fluoroquinolone-inhibited gyrase.

### 4.4. YacG

YacG is a 65-amino acid protein encoded within the genome of many proteobacteria [135], including *Escherichia coli*, and it has been identified as an endogenous inhibitor of DNA gyrase. The overexpression of YacG hampered *E. coli* cell growth by the inhibition of gyrase supercoiling and relaxation activities [136]. Furthermore, these studies established the specificity of YacG inhibition for gyrase with no inhibition of the related topoisomerase I or topoisomerase IV [136]. YacG directly interacts with the GyrB subunit with a tight affinity (K_Dapp_, 350 nM) [136]. Subsequently, the structure of YacG bound to a truncated version of *E. coli* gyrase was determined [137]. The previously characterized [138] zinc finger domain of YacG contacts the core region of the GyrB domain with a critical insertion of a YacG tryptophan side chain, while the C-terminal region of YacG is essential to interactions spanning GyrB and GyrA regions. This triggers an ordering of the adjacent QRDR (quinolone resistance-determining region) loop within GyrA, resulting in a steric blocking of the DNA-binding site of gyrase [137]. Typically, this QRDR loop is disordered in the absence of DNA but becomes structured upon YacG binding; furthermore, mutations within this loop result in fluoroquinolone resistance [137].

Subsequent studies clarified a highly protective role for YacG, and its expression appears to be induced by stresses that impact DNA gyrase integrity, including multiple classes of antibiotics that are less effective when YacG is present [135]. In bacterial cells containing the *yacG* gene, the oxidative stress induced by gyrase poisoning triggers an increase in the expression of YacG. This upregulated YacG then binds to gyrase, forming a gyrase–YacG complex that is incapable of binding to DNA (Figure 6). As a result, this complex formation prevents the subsequent steps leading to DNA damage and cell death, thereby protecting the cells from the action of gyrase poisons [135]. This is consistent with limiting spurious ATP turnover outside of catalysis, which is a previously noted consequence of YacG–gyrase interactions [136]. Interestingly, the site of YacG binding has been independently identified as a novel site for inhibitor binding through compound screening campaigns, and it continues to hold promise for the development of novel anti-gyrase therapeutic approaches [139].

### 4.5. MurI

Glutamate racemase (MurI) is an enzyme that plays an irreplaceable role in the early stages of peptidoglycan biosynthesis, converting L-glutamate to D-glutamate [140]. In addition to racemase activity, it was discovered that *E. coli* MurI also possesses a gyrase-inhibitory property (Figure 7) [141]. *E. coli* MurI inhibits DNA supercoiling in vivo and inhibits DNA gyrase in vitro in the presence of UDP-N-acetylmuramyl-L-alanine, which is a peptidoglycan precursor and MurI enzyme activator [141]. 

Subsequent studies observed similar inhibition mediated by the MurI enzyme (also named YrpC) from *Bacillus subtilis* [142], *Mycobacterium tuberculosis* [143], and *Mycobacterium smegmatis* [144]. Of note, the inhibition of gyrase by the *B. subtilis* enzyme is not dependent on MurI precursor substrates for activation, and it appears to inhibit gyrase even in the presence of the D-glutamate product [141]. The Mycobacterial versions of MurI inhibit both supercoiling and relaxation reactions of DNA gyrase [143]. It has been shown that MurI does not alter the activity of topoisomerase I [143], although no reports clarify if MurI can also regulate TopoIV. 

*M. tuberculosis* MurI directly bound to the GyrA subunit in far-Western blot experiments, and, using gel-based mobility shift assays, they were demonstrated to prevent the association of DNA with gyrase [143,144]. Furthermore, *M. tuberculosis* MurI can inhibit both *M. smegmatis* and *E. coli* gyrases, indicating that the interaction is not species-specific [143]. The gyrase inhibition by *M. smegmatis* MurI was found to protect against the action of ciprofloxacin, as measured by a decrease in the resulting gyrase–DNA covalent complex that arises from fluoroquinolone inhibition [144]. 

There are many structures of glutamate racemase from different bacterial species in the Protein Data Bank (for example, [145,146,147,148,149,150]); however, gyrase inhibition has only been published for the four MurI homologs described above. All structures of glutamate racemase contain a conserved active site bridged between two domains [140,148]. Currently, there is no information for the molecular mechanism of its interaction with DNA gyrase. The racemization reaction and gyrase inhibition by MurI appear to occur simultaneously but are not coupled, suggesting the “active” regions are distinct [144]. This potentially provides a regulating connection between D-glutamate levels, which are the product of MurI catalysis and are required for peptidoglycan synthesis, and the continuation and completion of cell replication (Figure 7).

### 4.6. GyrI

The gyrase inhibitory protein GyrI, also known as SbmC, is a chromosomally encoded *E. coli* gyrase regulator [151,152]. The *gyrI* gene is induced via the SOS response in the presence of DNA-damaging agents or when the cells enter the stationary phase [153]. Consistent with this, GyrI expression does not alter DNA topology but does offer protection from DNA-damaging agents such as mitomycin C [154,155]. Binding to gyrase was measured as approx. 30–40 µM to either GyrA or GyrB, but it strengthened to approx. 1.3 µM when the gyrase subunits were combined into a GyrAB complex [152]. GyrI inhibits both the supercoiling and relaxation activities of DNA gyrase, but no accumulation of dsDNA breaks occurs [152]. Instead, GyrI competes with gyrase for binding to DNA, and in this capacity, it can serve to protect gyrase from other types of inhibitors, including MccB17 and CcdB [152,154]. 

Recent work has identified a large family of mammalian proteins with embedded GyrI domains [156]. These proteins appear associated with the regulation of topoisomerase I, and for some members, it may link to DNA repair. There are evolutionary links between these mammalian proteins and bacterial and metazoan versions. The GyrI domain was found in tandem with different DNA binding domain motifs (MerR and AraC), with acetyltransferase domains, and with Hsp90 ATPase activator domains [156].

GyrI consists of two four-stranded antiparallel β-sheets and two α-helices [157]. A peptide corresponding to amino acids 89-ITGGQYAV-96 was able to effectively inhibit gyrase activity with an IC_50_ value of 12.3 µM and a binding affinity (K_D_) determined by the SPR of approx. 2.2 µM [151]. When this peptide sequence is mapped onto the structure of GyrI [157], it appears nestled in a beta sheet (Figure 8, yellow ribbon). An additional peptide corresponding to amino acids 33–48 (Figure 8, orange ribbon) also bound to gyrase but with a larger K_D_ value of 26 µM [152]. Interestingly, mammalian proteins containing GyrI domains do not conserve either of these peptide sequences, although other motifs are highly conserved [156]. Overall, it remains unclear how GyrI is able to effectively compete for DNA binding to gyrase.

## 5. Conclusions

DNA gyrase is a complex macromolecular machine, and its unique catalytic activity is essential to regulate the topology of circular double-stranded DNA. Gyrase is a valued antibacterial target with quinolone compounds serving as effective poisons leading to DNA breaks and cell death. As with many antibacterial targets, there are natural products that also interfere with functioning as a means of competition with neighboring cells or mobile genetic elements. Evolution would then favor cells that developed means of protecting these valued targets, like gyrase, culminating in a population of “foes” and “friends” with respect to molecular modes of action.

These differing activities can provide critical insight into fundamental mechanisms of gyrase catalysis as well as highlight weak points that may lead to the development of productive inhibitors. Given the cryptic nature of most of these binding sites, allosteric inhibition is likely important in regulating when gyrase is inhibited during catalysis, as demonstrated by compounds such as the thiophene class and evybactin [29,31]. In contrast, many molecules selectively compete with gyrase and related topoisomerases to block binding of the DNA substrates, effectively keeping gyrase activity sequestered to prevent aberrant ATP turnover and DNA breaks. As more molecular mechanisms become accessible, the field will benefit by increasing the breadth of inhibitor compound designs.

## Figures and Tables

**Figure 1 biology-13-00084-f001:**
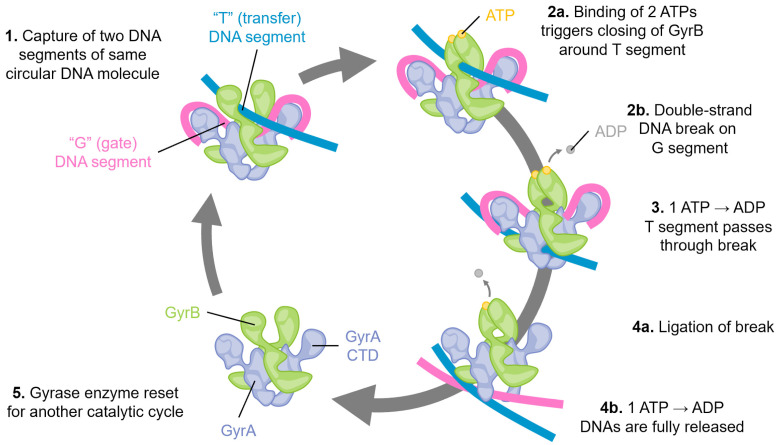
Schematic of the catalytic cycle of DNA gyrase. Step 1: Two segments of double-stranded DNA are captured on, for example, an underwound circular DNA molecule, and these are referred to as the “T”, or transfer segment, and the “G”, or gate segment. Note how the G segment is wrapped around the C-terminal domains (CTD) of GyrA. This unique arrangement is specific to gyrase enzymes. Step 2: a. The binding of two molecules of ATP locks the GyrB domains around the T segment. b. A double-stranded break is made in the G segment. This is mediated by the GyrA active site using a tyrosine amino acid, resulting in a covalent DNA-protein intermediate. Step 3: The hydrolysis of one ATP is coupled with large conformational changes, resulting in passing the T segment through the DNA break and into the central part of GyrA. Step 4: a. The DNA break is ligated, reforming the intact DNA molecule. b. Concomitant with ligation, hydrolysis of the second ATP molecule triggers complete DNA release, and Step 5: resets the gyrase enzyme for another catalytic cycle by triggering the re-opening of the GyrB domains.

**Figure 4 biology-13-00084-f004:**
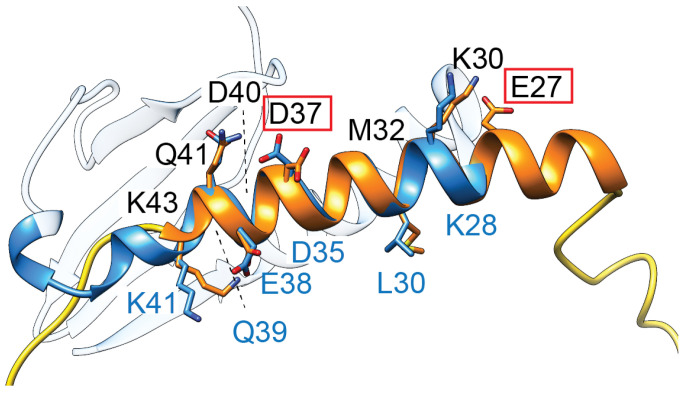
TsbT toxins are predicted to form a helical structure (orange) containing two residues (E27, D37, red boxes) critical for toxicity. This helix superposes relatively well with the second helix of ParE toxins (blue, amino acids denoted along the bottom in blue font), which has a relatively higher sequence conservation in the ParE protein family.

**Figure 5 biology-13-00084-f005:**
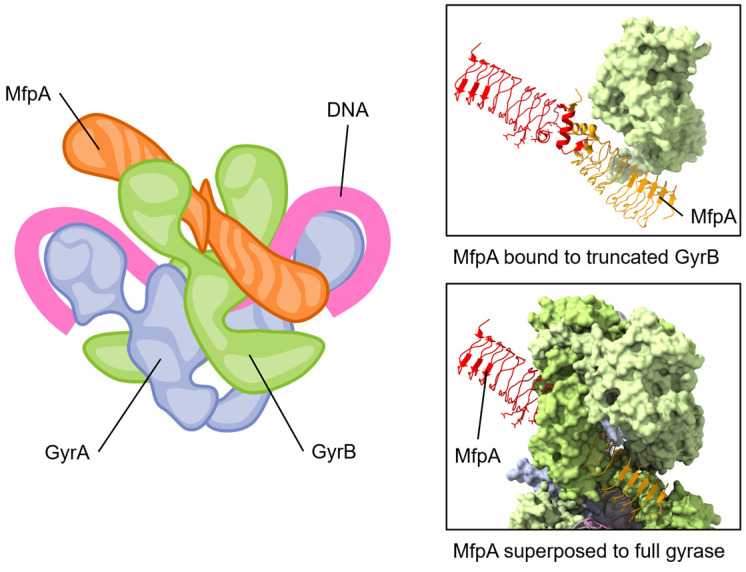
The pentapeptide repeat protein MfpA protein binds to GyrB and mimics DNA. The conserved dimeric solenoid structure and charge distribution (surface of MfpA, right) makes similar interactions as double-stranded DNA. The length of MfrP extends down GyrB (green, one GyrB hidden to allow viewing of MfpA) into the active site of GyrA (blue, gray), suggesting a mechanism for relieving fluoroquinolone poisoning of gyrase. Note that the “closed” form of GyrB in the complete gyrase structure has steric clashes with MfpA; for this reason, one GyrB chain has been removed from the depicted view. Superposed structures: PDB 6ZT5 MfpA with *M. smegmatis* GyrB N-terminal domain (only MfpA is shown here) [133]; PDB 6RKW, *E. coli* GyrA and GyrB bound to a 130 bp DNA [87].

**Figure 6 biology-13-00084-f006:**
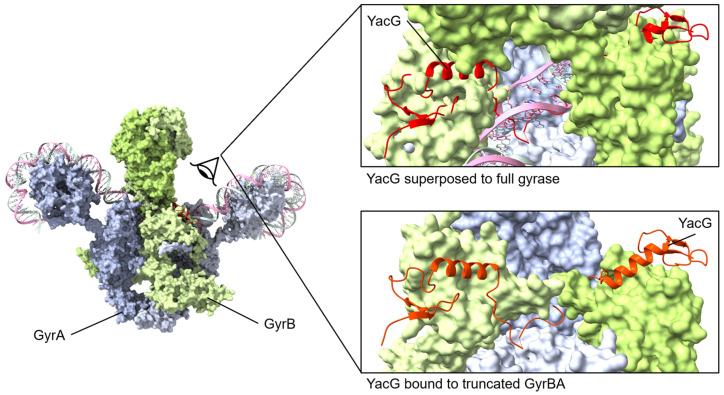
YacG protein binds to GyrB with the C-terminal region interacting with GyrB and GyrA regions. The small YacG protein inserts into openings among GyrB domains and extends into the central region adjacent to the GyrA active site. The tail of YacG can be visualized as likely clashing with DNA binding. *Superposed structures*: PDB 4TMA, YacG bound to a truncated *E. coli* GyrA GyrB [137], PDB 6RKW, *E. coli* GyrA and GyrB bound to a 130 bp DNA [87].

**Figure 7 biology-13-00084-f007:**
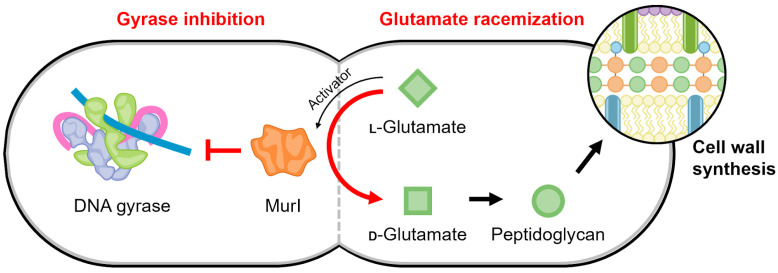
Schematic of the proposed regulatory link mediated by MurI to maintain a balance between gyrase activity and cell replication. The continued production of peptidoglycan precursors may be temporally regulated to occur when DNA replication is not active, such that the activity of gyrase is no longer needed.

**Figure 8 biology-13-00084-f008:**
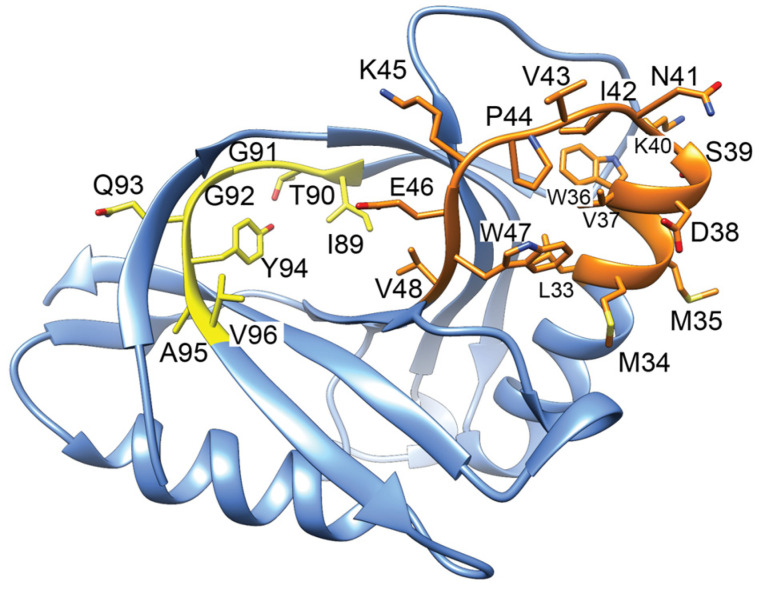
Structure of GyrI (PDB 1JYH, [157]) with corresponding gyrase-inhibiting peptides indicated. Peptides 89–96 (yellow) binds to gyrase approx. 10-fold more tightly than a peptide corresponding to 33–48 (orange). The yellow peptide is inhibitory when included in in vitro gyrase activity assays.

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
