# Peer review of "Friend or Foe: Protein Inhibitors of DNA Gyrase"

_biology, 2024, doi:10.3390/biology13020084_

Round 1

Reviewer 1 Report

Comments and Suggestions for Authors

This is a review of protein inhibitors of DNA gyrase, which is an essential enzyme in bacteria and an important target for antibacterial agents.  A review on this topic is useful and timely.  I found the article to be generally well-written, mostly factually correct (as far as I can tell) and was a pleasure to read.  As far as I could see the references were correctly cited.  I have few comments/suggestions for the authors:

Major comments:

1.     Lines 77-85: it might be valuable here to cite Osheroff’s work that suggests that gyrase can work much faster under some circumstances (Ashley, R.E., et al. (2017) Activities of gyrase and topoisomerase IV on positively supercoiled DNA. Nucleic Acids Res, 45, 9611-9624).

2.     In Figure 1, the substrates for gyrase are not necessarily ‘underwound’.

3.     Figure 1: what is the evidence for sequential ATP hydrolysis with gyrase?  I am only familiar with Lindsley’s work on yeast topo II.  (Also lines 138-139.)

Minor comments:

       i.          Legend to Fig. 3, line 231: better to say ‘GyrA dimers’ I think.

      ii.          Line 365: I think this should say: ‘the first and the third’.

     iii.          Line 391: space missing before [120].

    iv.          Line 409: should this say: ‘known to signal DNA damage’?

      v.          Line 460: should this say: ‘contacts the core region’?

Reviewer 2 Report

Comments and Suggestions for Authors

I found the manuscript interesting and am happy to recommend its publication.

My area of expertise is small molecule inhibitors of DNA gyrase and topoisomerase IV. Given the recent clinical successes of both zoliflodacin and gepotidacin - I would have liked this review on protein inhibitors of gyrase to include more on topoisomerase IV.

General comments:

1. What about topoisomerase IV? Please rephrase these two sentences for clarity.

 DNA gyrase is essential for successful replication of circular chromosomes, such as those found in most bacterial species. This is because it is the only enzyme that can relieve topological stressors associated with unwinding the double-stranded genetic material.

2. Mention only gyrase in M. tb,. This statement that ' is smaller and more compact than other gyrase enzyme' is not correct.

Correct this from:

 In contrast, the gyrase from Mycobacterium tuberculosis has relatively lower activity and structurally is smaller and more compact than other gyrase enzymes [35].

to something like:

In contrast, Mycobacterium tuberculosis only has DNA gyrase (it has no topoisomerase IV), has relatively lower activity and also lacks the Gram-negative specific insertion in GyrB. (ref. Caron and Wang, 1994?). The inhibition of the M. tb type I toposiomerase first, should lead to increased negative supercoiling by M. tb DNA gyrase, encouraging the bacteria out of the dormant state (Nagaraja, V. "Regulation of DNA topology in mycobacteria." Current Science (2004): 135-140. - or a more up to date reference).

3. Include or rephrase this interesting observation to include a reference to this paper:

(Gubaev, A., Weidlich, D., & Klostermeier, D. (2016). DNA gyrase with a single catalytic tyrosine can catalyze DNA supercoiling by a nicking-closing mechanism. Nucleic acids research, 44(21), 10354-10366.)

Your observations suggest that it is the unique negative supercoiling reaction of DNA gyrase that is inhibited - which can be achieved by a different mechanism.

ParE toxins do not inhibit topoisomerase IV, nor the relaxation reactions mediated by gyrase ([52,53] and unpublished observation). The inhibition of gyrase by ParE toxins requires ATP hydrolysis [52,53], indicating that ParE toxins likely recognize and stabilize a catalytic intermediate adopted after dsDNA cleavage

4. Please correct typo on page 13:

' are required åfor peptidoglycan synthesis'

5. Conclusions. Consider this citation for inclusion in discussion of allostery:

Imai, Yu, et al. "Evybactin is a DNA gyrase inhibitor that selectively kills Mycobacterium tuberculosis." Nature chemical biology 18.11 (2022): 1236-1244.
